# Prevalence of dengue in febrile patients in Peru: A systematic review and meta-analysis

**Darwin A. León-Figueroa**[1,2], **Edwin A. Garcia-Vasquez**[1], **Milagros Diaz-Torres**[1], **Edwin Aguirre-Milachay**[3], **Jean Pierre Villanueva-De La Cruz**[4], **Hortencia M. Saldaña-Cumpa**[1], **Mario J. Valladares-Garrido**[5*]

**1** Facultad de Medicina Humana, Universidad de San Martín de Porres, Chiclayo, Peru, **2** Hospital Nacional Sergio E. Bernales, Lima, Perú, **3** Hospital Nacional Almanzor Aguinaga Asenjo, EsSalud, Chiclayo, Peru, **4** Facultad de Medicina de la Universidad Nacional de Trujillo, La Libertad, Perú, **5** Escuela de Medicina Humana, Universidad Señor de Sipán, Chiclayo, Peru

\* vgarrido@uss.edu.pe

## Abstract

### Background

Dengue is an acute febrile illness that is a significant public health problem. Peru is an endemic region for vector-borne diseases such as dengue, zika, and chikungunya, which initially manifest with febrile illness and can complicate differential diagnosis. Therefore, the present study aimed to determine the prevalence of positive results for dengue or dengue antibodies in Peruvian patients with febrile illness using diagnostic tools such as RT-PCR and ELISA NS1, IgM, and IgG.

### Methods

A literature search was conducted in eight databases or search tools (PubMed, Scopus, Embase, Web of Science, ScienceDirect, Google Scholar, Virtual Health Library, and Scielo) until June 9, 2024. Medical Subject Headings (MeSH) terms such as "dengue" and "Peru" were used, together with the free term "febrile illness", combined using the Boolean operators AND and OR. We included observational studies with a control group of patients with fever but no dengue infection and a non-control group of febrile patients who tested positive for dengue. Pooled estimates and 95% confidence intervals (CI) were calculated using random-effects models. Study quality and risk of bias were assessed using the Joanna Briggs Institute Statistical Meta-Analysis Assessment and Review Instrument. Heterogeneity was assessed using the $I^2$ statistic, and statistical analysis was performed with R version 4.2.3.

### Results

We included 15 observational studies that met the inclusion criteria developed in 10 regions of Peru and published between 2002 and 2022, with a total of 12,355

**Data availability statement:** All relevant data are within the manuscript and its Supporting information files.

**Funding:** The author(s) received no specific funding for this work.

**Competing interests:** The authors have declared that no competing interests exist.

patients with febrile illness. The pooled prevalence of positive results for dengue or dengue antibodies in these patients was 21% (95% CI: 9%−36%; 2022 participants; 5 studies; $I^2 = 98\%$) for IgG ELISA, 16% (95% CI: 11%−21%; 10891 participants; 10 studies; $I^2 = 97\%$) for IgM ELISA, 19% (95% CI: 9%−31%; 2086 participants; 5 studies; $I^2 = 98\%$) for NS1 ELISA, and 20% (95% CI: 13%−28%; 3107 participants; 9 studies; $I^2 = 96\%$) for RNA PCR.

## Conclusion

Our results suggest a high prevalence of positive results for dengue or dengue antibodies among febrile patients in Peru, which varies depending on the diagnostic method used. Despite this variability, the use of accurate diagnostic methods is essential for the early detection and prevention of serious complications. The findings underline the need to strengthen dengue control strategies, improve diagnostic capacity in health centers, and optimize epidemiological surveillance in high-incidence regions. It is recommended that public health policies focus on these key areas for better management of the disease.

---

## 1. Introduction

Fever is one of the main reasons why patients in tropical areas seek medical attention. Although it often presents with non-specific gastrointestinal or respiratory symptoms, it can also manifest in isolation [1,2]. In this context, the term "fever" is commonly used as a synonym for "febrile illness," which is defined by a body temperature of 38.0°C or higher [1,3]. The etiological diagnosis of acute febrile illness is complex, as different diseases can present similarly, especially in the early stages [4,5], highlighting the need for more precise diagnostic methods [4,6].

Dengue stands out as a significant viral infection and a leading cause of undifferentiated febrile illness [7]. The dengue virus, transmitted by Aedes mosquitoes, is a positive-stranded RNA virus that can be fatal and is a major public health issue worldwide, especially in tropical and subtropical countries [8]. There are four viral serotypes: DENV-1, DENV-2, DENV-3, and DENV-4 [9]. In 2024, more than 13 million cases of dengue have been reported across North, Central, and South America, as well as the Caribbean, with Peru reporting 280,726 dengue cases and 259 related deaths, ranking sixth in the Americas in terms of incidence after Brazil, Argentina, Mexico, Colombia, and Paraguay [10,11]. As of epidemiological week 14 in 2025, Peru reported 26,597 cases and 31 related deaths [12].

Most people infected with dengue present with mild or no symptoms, which usually last between 2 and 7 days [13,14]. A systematic review reported that the most common symptoms in Peruvian patients with dengue included general malaise, fever, headache, arthralgia, myalgia, retroocular pain, lumbar pain, and rash/exanthema [13]. A crucial aspect for the effective management of dengue is accurate and timely diagnosis, as its non-specific clinical manifestations can be similar to those of other

mosquito-borne viral diseases such as yellow fever, chikungunya, and Zika, as well as other diseases like malaria, trypanosomiasis, leptospirosis, and Chagas disease [15,16].

The initial diagnosis of dengue is based on clinical assessment, followed by laboratory testing. These tests include direct methods such as viral isolation by culture, nucleic acid detection by conventional PCR or qRT-PCR (quantitative Real Time-Polymerase Chain Reaction), and NS1 (non-structural protein 1) antigen identification. In addition, indirect methods, such as detection of IgM and IgG antibodies by enzyme-linked immunosorbent assays (ELISA), are used to distinguish between recent and past exposure to the virus [17]. Throughout their lives, people can become infected with different serotypes of dengue virus. Patients who have already been infected by one serotype and then contract another often develop more severe disease due to the antibody-dependent enhancement phenomenon, in which antibodies generated by the first infection facilitate replication of the new virus [18].

Although viral isolation and real-time PCR are standard methods, they require specialized infrastructure and trained personnel. In contrast, tests such as NS1 and IgM/IgG ELISA are more accessible and rapid, making them ideal for resource-limited settings [19,20]. Therefore, this study aims to determine the prevalence of positive results for dengue or dengue antibodies in Peruvian patients with febrile illness using diagnostic tools such as RT-PCR and ELISA for NS1, IgM, and IgG. The study aims to provide detailed information on the accuracy of these methods in dengue detection, which will improve the diagnosis and management of the disease in the region.

## 2. Materials and methods

### 2.1. Protocol and registration

The current investigation followed the Preferred Reporting Items for Systematic Reviews and Meta-Analyses (PRISMA) guidelines [21] (S1 Table) as well as a protocol registered in the Prospective International Registry of Systematic Reviews (PROSPERO) with the number CRD42024558891. The original protocol underwent minor adjustments, which are detailed as follows: (1) the inclusion criteria were established, defining a control group and a non-control group; (2) the list of authors was updated to reflect changes in team participation during the development of the review; and (3) additional researchers were incorporated into the processes of article screening, evaluation, and data analysis to ensure greater rigor and reduce the risk of bias through independent assessments. These modifications were implemented prior to the analysis stage and did not alter the main objective or methodology of the review.

### 2.2. Eligibility criteria

This review included studies reporting the prevalence of dengue virus infection in Peruvian patients with febrile illness. Study participants were of both sexes and any age range. The investigation included a control group, consisting of patients with fever but without dengue infection, and a non-control group, consisting of febrile patients who tested positive in any or all diagnostic tests for dengue. Observational studies were considered, including retrospective, prospective, cohort, and cross-sectional studies. We included studies that used diagnostic tools like RT-PCR and ELISA (NS1, IgM, and IgG) to diagnose dengue. We excluded studies that did not meet the established criteria, including case reports, editorials, letters to the editor, randomized clinical trials, conference proceedings, and narrative or systematic reviews.

### 2.3. Information sources and search strategy

We searched eight databases or search tools, including PubMed, Scopus, Embase, Web of Science, ScienceDirect, Google Scholar, Virtual Health Library, and Scielo, without any restrictions on language or development period, until June 9, 2024. Medical Subject Headings (MeSH) terms such as "dengue" and "Peru" were used, together with the free term "febrile illness", combined using the Boolean operators AND and OR. S2 Table details the search strategy that two authors independently validated. We also used other search methods, such as reviewing literature studies, consulting article

references, and scanning publications in Peruvian journals that specialize in infectious and communicable diseases. However, the potential studies identified were within the scope of the search strategy employed.

## 2.4. Study selection

The reference manager EndNote version X9 (Thomas Reuters, New York, NY, USA) stored the search strategy's results. We then removed the duplicate articles. Three researchers independently reviewed the titles and abstracts to determine their eligibility based on pre-established selection criteria. Finally, we conducted a thorough review of the full articles to determine if they met the inclusion criteria. A fourth investigator intervened to resolve discrepancies between reviewers at each stage of the selection process.

## 2.5. Outcomes

The main objective is to determine the prevalence of positive results for dengue or dengue antibodies in Peruvian patients with febrile illness using diagnostic tools such as RT-PCR and ELISA NS1, IgM, and IgG.

## 2.6. Quality assessment

We used the JBI-MAStARI (Joanna Briggs Institute Meta-Analysis of Statistics Assessment and Review Instrument) tool to assess the quality and risk of bias of the articles included in the meta-analysis. This assessment covered several aspects, such as study context, outcomes and explanatory variables, specific inclusion criteria, measurement methods applied, a detailed description of the topic, and a comprehensive statistical analysis. We answered each question with either "yes", "no", "unclear", or "not applicable". The quality of the studies was classified as high (≥ 7 points), moderate (4–6 points), or low (< 4 points) according to their scores (S3 Table) [22].

## 2.7. Data collection process and data items

We compiled the data from the articles into an Excel spreadsheet. Three authors separately and manually gathered different types of information, such as the author's name, the year the study was published, the study design, the number of patients with febrile illness, the geographical region, the sex of the patients (male or female), the diagnostic methods used (RT-PCR and ELISA), the biomarkers found (RNA, NS1, IgM, and IgG), and the target population (suspected dengue cases or patients with acute fever). After individual extraction, the three authors met to review their findings and identify any discrepancies. Consensus was achieved through structured discussion, where each author presented their rationale for the extracted data. In cases of disagreement, the team referred back to the original full text of the article to collectively determine the most accurate interpretation. Then, to ensure the accuracy and quality of the extracted data, a fourth independent researcher conducted a thorough review and verification.

## 2.8. Data analysis

We performed a prevalence (proportions) meta-analysis using R software version 4.2.3. We used a weighted inverse variance random effects model to find out how common dengue was among Peruvians with febrile illness. We looked at diagnostic tools like RT-PCR and ELISA NS1, IgM, and IgG (S6 Table). To assess between-study variability, we applied the Cochrane Q statistic. Between-study heterogeneity was measured using the inconsistency index ($I^2$), classifying it as follows: low if the $I^2$ value was less than 25%, moderate if it was between 25% and 50%, and high if it exceeded 75%. $I^2$ values between 50% and 75% were considered indicative of substantial heterogeneity [23].

We used two methods to assess potential publication bias: visual inspection of the funnel plot and Egger's test. We only applied these methods to diagnostic tools that included at least 10 studies in the meta-analysis, as the test's power to detect true asymmetry with fewer studies is limited. A bias in the results was considered to be present when the p-value was less than 0.05 [24].

The results of the research were presented in tables and descriptive graphs. The combined prevalence of positive results for dengue or dengue antibodies in Peruvian patients with febrile illness using diagnostic tools such as RT-PCR and ELISA NS1, IgM, and IgG was shown graphically with a forest plot, which included 95% confidence intervals to provide a more accurate presentation of the data.

## 3. Results

### 3.1. Study selection

Through searches of eight databases or search tools, 1,390 studies were identified. After removing duplicates (n = 800), the titles and abstracts of the remaining 590 studies were reviewed. Subsequently, 49 full-text articles were assessed, of which 15 met the inclusion criteria for the systematic review and meta-analysis (S4 Table). Fig 1 presents the PRISMA flow chart, which illustrates the selection process [17,19,25–37].

### 3.2. Characteristics of the included studie

An analysis was performed based on a review of 15 observational studies published between 2002 and 2022 that investigated the prevalence of positive results for dengue or dengue antibodies in Peruvian patients with febrile illness, using diagnostic tools such as RT-PCR and ELISA for NS1, IgM, and IgG. In total, 12,355 patients with febrile illness were included. The studies were conducted in 10 regions of Peru, with a main focus on Piura and Cajamarca (Table 1) [17,19,25–37].

### 3.3. Quality of the included studies and publication bias

The quality of the studies included in the analysis was moderate, as shown in S3 Table [17,19,25–37]. In the analysis assessing the pooled prevalence of positive results for dengue or dengue antibodies in Peruvian patients with febrile illness by IgM ELISA, a forest plot asymmetry was evident. In addition, Egger's test yielded a value of p = 0.0121 (t = −3.23, df = 8), suggesting a possible publication bias (S1 Fig) [17,19,25,29,30,33–37].

### 3.4. Prevalence of positive results for dengue or dengue antibodies in Peruvian patients with febrile illness according to NS1, IgM, IgG ELISA, and RT-PCR

An analysis of the prevalence of positive results for dengue or dengue antibodies in Peruvian patients with febrile illness was performed using diagnostic tools such as RT-PCR and ELISA for NS1, IgM, and IgG, according to the data in S5 Table [17,19,25–37]. The pooled prevalence of positive results for dengue or dengue antibodies in these patients was 21% (95% CI: 9–36%; 2022 participants; 5 studies; $I^2$ = 98%, p < 0.01) for IgG ELISA (Fig 2) [17,19,29,30,35], 16% (95% CI: 11–21%; 10,891 participants; 10 studies; $I^2$ = 97%, p < 0.01) for IgM ELISA (Fig 3) [17,19,25,29,30,33–37], 19% (95% CI: 09–31%; 2086 participants; 5 studies; $I^2$ = 98%, p < 0.01) for NS1 ELISA (Fig 4) [17,19,25,29,30], and 20% (95% CI: 13–28%; 3107 participants; 9 studies; $I^2$ = 96%, p < 0.01) for RNA PCR (Fig 5) [19,25–29,31–33].

## 4. Discussion

### 4.1. Main findings

In this study, we conducted a systematic review and meta-analysis to determine the prevalence of positive results for dengue or dengue antibodies in Peruvian patients with febrile illness using diagnostic tools such as RT-PCR and ELISA (NS1, IgM, and IgG). Our results indicate a pooled prevalence of dengue antibodies (IgG and IgM) and viral markers (NS1 and RNA) of 21% for IgG ELISA, 16% for IgM ELISA, 19% for NS1 ELISA, and 20% for RNA PCR.

Our findings align with previous studies that have reported a variable prevalence of dengue in endemic regions. Rosenberger KD, et al. reported in their study on 5189 patients with febrile illness in eight countries in Asia and Latin America.

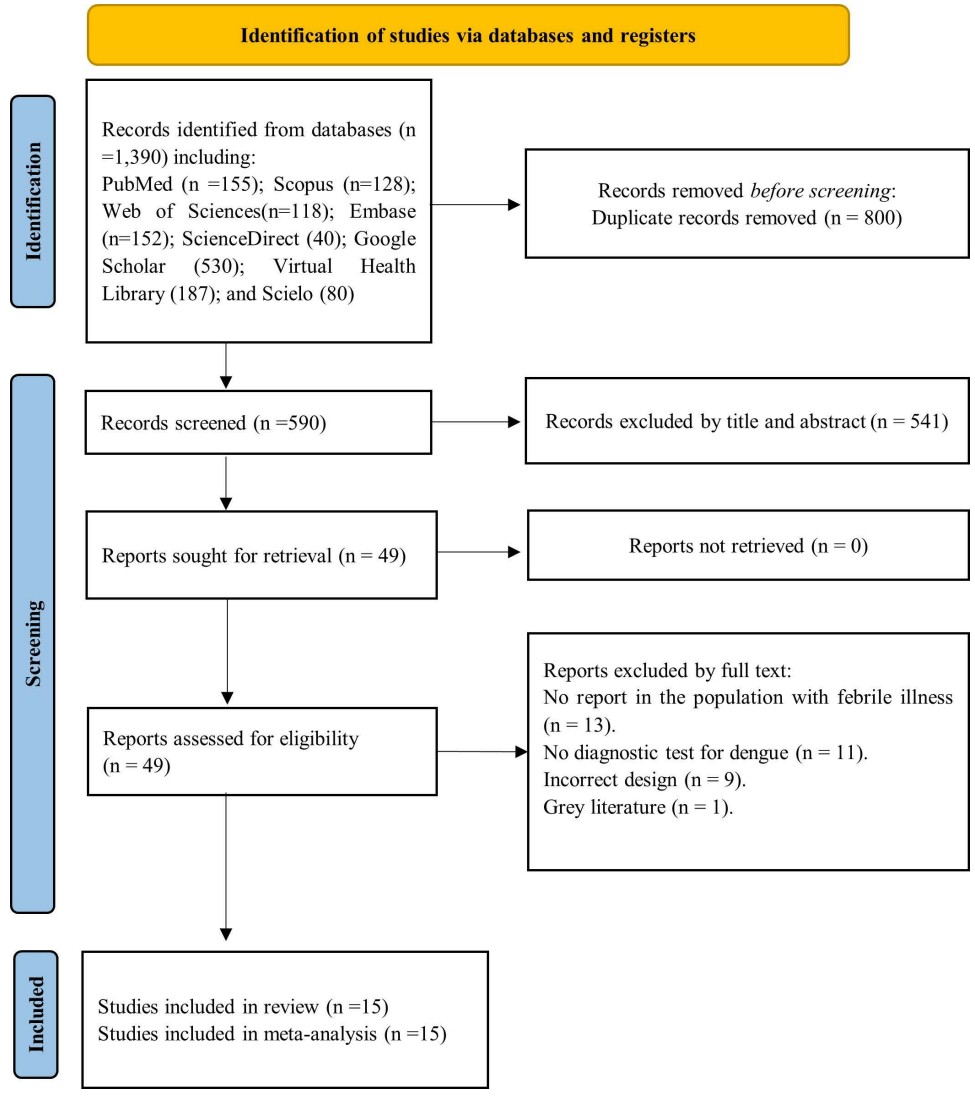

**Fig 1. Study selection process based on the PRISMA flowchart.**

Of these, 2694 (36%) were diagnosed with laboratory-confirmed dengue, while 2495 (34%) had other febrile illnesses unrelated to dengue [38]. Simo FBN et al., in their systematic review and meta-analysis study, found that the prevalence of dengue in African resident fever populations was 24.8% for IgG (21 studies, 7050 patients), 10.8% for IgM (36 studies, 40251 patients), and 7.1% for RNA (21 studies, 15322 patients) [39]. In India, Sarkar S. et al. reported that of 316 samples from patients with febrile illness, the prevalence of dengue was 32.5% (103) that were reactive for the NS1 antigen [40]. A meta-analysis by Nigussie E et al. in Ethiopia, which included patients with suspected dengue and febrile illness, estimated a pooled prevalence of dengue of 21% (4 studies, 1848 patients) for IgG, 9% (5 studies, 2255 patients) for IgM, and 48% (6 studies, 258 patients) for RNA [41].

It is important to consider that dengue exhibits both endemic and epidemic patterns, with cyclical outbreaks and fluctuating incidence throughout the year. These seasonal and interannual variations can lead to spikes in cases during specific periods, particularly during rainy seasons or climatic events such as El Niño. As a result, interpreting and comparing

**Table 1. Summary of studies on dengue in Peruvian patients with febrile illness included in the systematic review and meta-analysis.**

| Authors | Year | Studio Type | Patients with febrile illness | Geographical region | Dengue test | Biological markers | | | | Population |
|---|---|---|---|---|---|---|---|---|---|---|
| | | | | | | IgG | IgM | RNA | NS1 | |
| Valdivia-Conroy B, et al. [17] | 2022 | Observational | 286 | Lima | ELISA | 156 (54.5%) | 54 (19%) | NR | 97 (34%) | Febrile illness |
| arazona-Castro Y, et al. [25] | 2022 | Cross-sectional | 464 | Cajamarca | RT-PCR ELISA | NR | 43 (9.3%) | 21 (4.5%) | 21 (4.5%) | Febrile illness |
| Del Valle-Mendoza J, et al. [26] | 2021 | Cross-sectional | 276 | Piura | RT-PCR | NR | NR | 84 (30.4%) | NR | Febrile illness |
| Aguilar-Luis MA, et al. [19] | 2021 | Retrospective cohort | 359 | Cajamarca | RT-PCR ELISA | 60 (16.7%) | 35 (9.7%) | 89 (24.7%) | 109 (30.3%) | Febrile illness |
| Elson WH, et al. [27] | 2020 | Retrospective cohort | 429 | Loreto | RT-PCR | NR | NR | 79 (18.4%) | NR | Febrile illness |
| Del Valle-Mendoza J, et al. [28] | 2020 | Cross-sectional | 124 | Cajamarca | RT-PCR | NR | NR | 32 (25.8%) | NR | Febrile illness |
| Palomares-Reyes C, et al. [29] | 2019 | Retrospective cohort | 268 | Huánuco | RT-PCR ELISA | 42 (15.67%) | 28 (10.45%) | 69 (25.75%) | 51 (19.03%) | Febrile illness |
| Torres – Coronado PE, et al. [30] | 2019 | Retrospective cohort | 709 | Lambayeque | ELISA | 61 (8.6%) | 136 (19.2%) | NR | 108 (15.2%) | Probable case of dengue ⁂ |
| Sánchez-Carbonel J, et al. [31] | 2018 | Cross-sectional | 496 | Piura | RT-PCR | NR | NR | 170 (34.3%) | NR | Febrile illness |
| Alva-Urcia C, et al. [32] | 2017 | Cross-sectional | 139 | Madre de Dios | RT-PCR | NR | NR | 9 (6.5%) | NR | Febrile illness |
| Loayza M, et al. [33] | 2010 | Cross-sectional | 552 | Lima | RT-PCR ELISA | NR | 148 (26.8%) | 99 (17.9%) | NR | Suspect case * |
| Troyes RL, et al. [34] | 2006 | Retrospective cohort | 1039 | Cajamarca | ELISA | NR | 105 (10.1%) | NR | NR | Febrile illness |
| Gómez B, et al. [35] | 2005 | Cross-sectional | 400 | Áncash | ELISA | 61 (15.3%) | 40 (10%) | NR | NR | Probable case of dengue ⁂ |
| Cobos Z, et al. [36] | 2004 | Retrospective cohort | 742 | Ucayali | ELISA | NR | 142 (19.1%) | NR | NR | Probable case of dengue ⁂ |
| Mostorino ER, et al. [37] | 2002 | Retrospective cohort | 6072 | Lima | ELISA | NR | 1593 (26.2%) | NR | NR | Probable case of dengue ⁂ |

NR: Not Reported; ELISA: enzyme-linked immunosorbent assay; RT PCR: reverse transcriptase polymerase chain reaction; NS1-RDT: NS1-rapid diagnostic test; ⁂Probable case of dengue: febrile illness <7 days plus clinical manifestations; and *Suspect case: febrile illness <7 days.

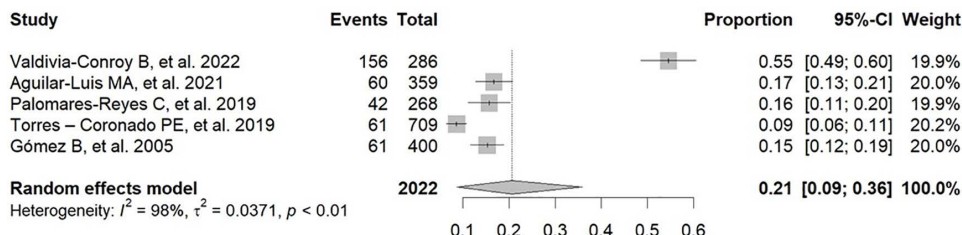

**Fig 2. Pooled prevalence of positive results for dengue or dengue antibodies in Peruvian patients with febrile illness by IgG ELISA.**

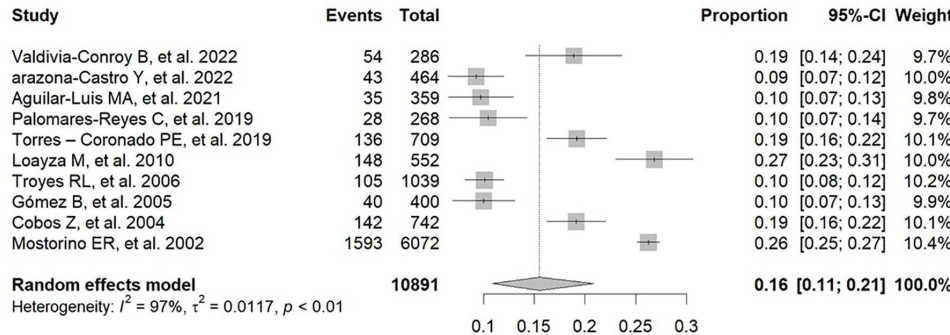

**Fig 3. Pooled prevalence of positive results for dengue or dengue antibodies in Peruvian patients with febrile illness by IgM ELISA.**

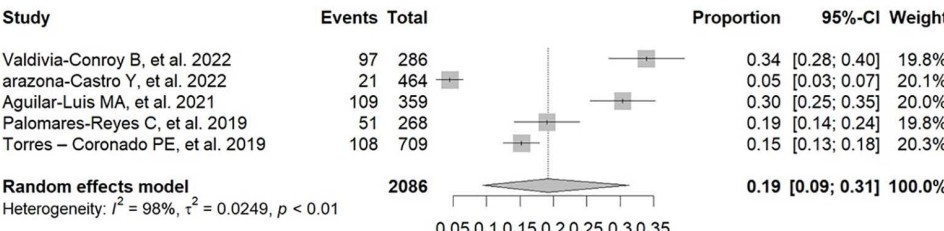

**Fig 4. Pooled prevalence of positive results for dengue or dengue antibodies in Peruvian patients with febrile illness by NS1 ELISA.**

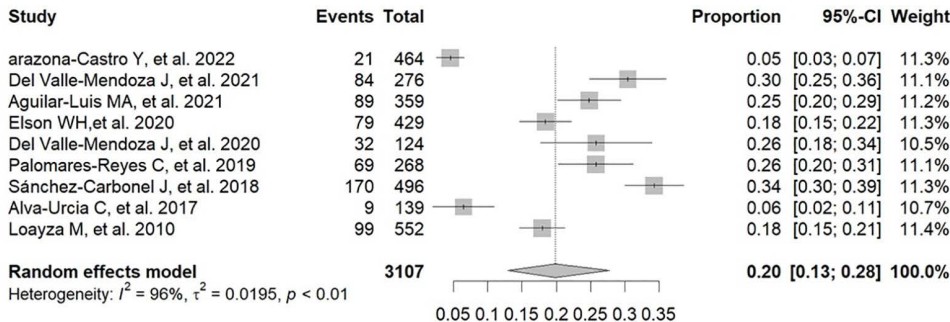

**Fig 5. Pooled prevalence of positive results for dengue or dengue antibodies in Peruvian patients with febrile illness by RNA PCR.**

prevalence estimates across studies conducted at different times and locations is challenging, as positivity rates may reflect temporary epidemic surges rather than baseline endemic transmission [42,43].

This discrepancy in prevalence rates could be due to a variety of factors. One of the main ones is heterogeneity in the diagnostic methods employed, which may affect both detection accuracy and case reporting [44]. Diagnostic strategies, adapted according to the resources and skills available in each laboratory, vary significantly, which introduces an additional source of variability [45]. In addition, demographic characteristics of the populations, such as population density, access to health services, and mobility of individuals, may also have contributed to the observed differences [46–48]. Finally, geographical variations, including climate and environmental conditions specific to the regions studied, may influence virus transmission, which would help explain disparities in dengue prevalence between different areas [49,50].

## 4.2. Public health implications of the findings

The results of this study, which reveal a high prevalence of positive results for dengue or dengue antibodies among febrile patients, underscore the need to implement a more comprehensive diagnostic approach in the different regions of Peru. In particular, our findings suggest the importance of using diagnostic tests with high sensitivity to detect acute infections, such as NS1 antigen detection tests, which can identify the virus in the early stages of infection. We also recommend the use of molecular tests, such as PCR, to improve diagnostic accuracy, especially in areas with high circulation of multiple dengue serotypes. Evaluating the sensitivity and specificity of these tools in various regions will help to optimize their use in local contexts. Particularly, the northern regions of the country, such as Piura and Lambayeque, play a crucial role due to meteorological conditions such as the coastal El Niño phenomenon. In addition, the lack of a strong and effective health system, political problems, and rapid unplanned urbanization contribute significantly to the high prevalence of dengue in these areas [47,51,52].

In this context, it is essential to maintain robust epidemiological surveillance and strengthen diagnostic capacities in these areas. Government efforts, including surveillance by agencies like the Centers for Disease Control and Prevention (CDC) and the Ministry of Health (MINSA), are vital in controlling outbreaks. Targeted surveillance during peak transmission periods, such as the rainy season, ensures timely interventions. Strengthening diagnostic infrastructure and integrating advanced surveillance systems enhance early detection and response to future outbreaks. It is imperative that health professionals in these regions maintain a high level of clinical suspicion for possible dengue cases, even when they present with atypical clinical manifestations, to ensure early diagnosis and appropriate therapeutic management [20,53]. In addition, other endemic diseases in Peru, such as leptospirosis, Zika, chikungunya, malaria, and others that cause febrile illnesses, must be considered. The COVID-19 pandemic has had significant implications for the surveillance and management of dengue. The shift in healthcare resources toward COVID-19 responses led to delays in routine dengue monitoring and a reduction in the capacity for vector control in some regions. These factors may have led to an underreporting of cases or delayed detection, which could impact the accuracy of prevalence estimates in the period during and after the pandemic [54,55]. It is therefore essential to implement clinical strategies that improve diagnostic capacity, including the use of rapid tests and ongoing training of medical staff for the accurate identification of dengue, even in its less common presentations [20,56].

## 4.3. Study quality and publication bias

An analysis of the quality of the studies included in our meta-analysis revealed that most were of moderate quality. The presence of publication bias, as indicated by Egger's test (p = 0.0121), and the asymmetry observed in the funnel plots might have influenced our prevalence estimates. This influence may be related to the geographical distribution of the populations in the included studies, which limits the generalizability of the conclusions. In addition, the diagnostic tools used in the studies, such as ELISA and PCR, may have been inconsistent in their preparation, execution, and interpretation, making accurate diagnosis difficult [38,57]. This underlines the need to apply rigorous controls and complementary methods to ensure the accuracy of the results. Finally, to mitigate publication bias in future studies, it is essential to use statistical techniques such as trim and fill, which adjust estimates for publication bias, as well as Bayesian methods and sensitivity analysis to assess the impact of publication bias on the conclusions of the meta-analysis [24,58,59].

## 4.4. Limitations and strengths

This study has several important limitations that should be considered when interpreting the results. First, high heterogeneity was observed in most analyses (I² = 96%–98%), which may limit the generalizability of our findings. This heterogeneity could be attributed to differences in study designs, diagnostic methodologies used, and the characteristics of the populations studied. Variability in diagnostic methods, such as RT-PCR and ELISA for NS1, IgM, and IgG, may also have influenced the accuracy and comparability of results, as these tools may vary in their sensitivity and specificity.

In addition, the moderate quality of the included studies suggests potential limitations in their design, execution, or reporting, which may have impacted the robustness and reliability of our conclusions. While the JBI-MAStARI tool is useful for assessing bias in individual studies, it has limitations, as it does not directly evaluate the feasibility of a meta-analysis or account for heterogeneity. Therefore, to achieve a more comprehensive evaluation, we used additional methods that addressed both publication bias and heterogeneity. We also identified a possible publication bias, as evidenced by Egger's test (p = 0.0121) and the asymmetry observed in the funnel plots. This suggests that studies with negative or nonsignificant results may have been less likely to be published, which could have influenced our prevalence estimates. In addition, we recognize the limitations of serologic testing to determine the prevalence of acute infection in febrile patients, especially in dengue-endemic communities. The detection of antibodies, such as IgG and IgM, may reflect past exposures to other pathogens, or in the case of IgG, previous dengue infections, which complicates the interpretation of results and may not represent the current cause of fever. To minimize these limitations and to obtain a more accurate picture of the current clinical situation, a variety of complementary diagnostic tests were employed [17,19,25–37].

Despite these limitations, the study also presents several notable strengths. First, the systematic review and meta-analysis included a wide range of observational studies conducted in various regions of Peru, providing a comprehensive view of the prevalence of dengue in patients with febrile illness. Additionally, the inclusion of robust diagnostic tools, such as RT-PCR and ELISA for NS1, IgM, and IgG, allowed for a comprehensive assessment of dengue prevalence from different diagnostic perspectives, enriching the interpretation of the results. Furthermore, the rigorous process of study selection, quality assessment, and data extraction, involving multiple investigators, ensures the accuracy and completeness of the analysis. Finally, the detailed presentation of the results through the application of advanced statistical methods provides a clear and complete representation of the findings, facilitating their understanding and application.

## 5. Conclusions

The results show an overall prevalence of positive dengue or dengue antibody findings of 21% for IgG ELISA, 16% for IgM ELISA, 19% for NS1 ELISA, and 20% for RT-PCR. These figures indicate a significant disease burden in the country, with variations in prevalence depending on the diagnostic tool used. The high prevalence observed, especially in regions such as Piura, highlights the need to improve diagnostic and disease management strategies in these areas. The use of multiple diagnostic tools can facilitate earlier and more accurate detection of dengue, which is crucial for the implementation of effective therapeutic measures and for the prevention of serious complications. Our findings also highlight the importance of strengthening dengue control and prevention strategies in high-incidence regions. It is recommended that public health policies focus on improving diagnostic capacity in health centers, strengthening prevention awareness campaigns, and optimizing epidemiological surveillance.

## Supporting information

**S1 Table. PRISMA Checklist (PRISMA 2020 Main Checklist and PRIMSA Abstract Checklist).** (DOCX)

**S2 Table. The adjusted search terms as per searched electronic databases or search tools.** (DOCX)

**S3 Table. Quality of the studies included in the systematic review and meta-analysis.** (DOCX)

**S4 Table. Table of studies evaluated for inclusion and exclusion.** (XLSX)

**S5 Table. Meta-analysis database.**

(DOCX)

**S6 Table. R version 4.2.3. script.**
(DOCX)

**S1 Fig. The funnel plot and Egger's test illustrate the publication bias of the included studies assessing the pooled prevalence of positive results for dengue or dengue antibodies in Peruvian patients with febrile illness by IgM ELISA.**
(TIF)

## Author contributions

**Conceptualization:** Darwin A. León-Figueroa, Edwin A. Garcia-Vasquez, Milagros Diaz-Torres, Edwin Aguirre-Milachay, Jean Pierre Villanueva-De La Cruz, Hortencia M. Saldaña-Cumpa, Mario J. Valladares-Garrido.

**Data curation:** Darwin A. León-Figueroa, Milagros Diaz-Torres, Edwin Aguirre-Milachay, Jean Pierre Villanueva-De La Cruz, Mario J. Valladares-Garrido.

**Formal analysis:** Darwin A. León-Figueroa, Edwin Aguirre-Milachay, Mario J. Valladares-Garrido.

**Investigation:** Darwin A. León-Figueroa, Edwin A. Garcia-Vasquez, Milagros Diaz-Torres, Edwin Aguirre-Milachay, Jean Pierre Villanueva-De La Cruz, Hortencia M. Saldaña-Cumpa, Mario J. Valladares-Garrido.

**Methodology:** Darwin A. León-Figueroa, Milagros Diaz-Torres, Edwin Aguirre-Milachay, Mario J. Valladares-Garrido.

**Project administration:** Darwin A. León-Figueroa, Edwin A. Garcia-Vasquez, Milagros Diaz-Torres, Edwin Aguirre-Milachay, Jean Pierre Villanueva-De La Cruz, Hortencia M. Saldaña-Cumpa, Mario J. Valladares-Garrido.

**Resources:** Darwin A. León-Figueroa, Edwin A. Garcia-Vasquez, Milagros Diaz-Torres, Edwin Aguirre-Milachay, Jean Pierre Villanueva-De La Cruz, Hortencia M. Saldaña-Cumpa, Mario J. Valladares-Garrido.

**Software:** Darwin A. León-Figueroa, Mario J. Valladares-Garrido.

**Supervision:** Darwin A. León-Figueroa, Milagros Diaz-Torres, Edwin Aguirre-Milachay, Jean Pierre Villanueva-De La Cruz, Hortencia M. Saldaña-Cumpa, Mario J. Valladares-Garrido.

**Validation:** Darwin A. León-Figueroa, Edwin A. Garcia-Vasquez, Milagros Diaz-Torres, Edwin Aguirre-Milachay, Jean Pierre Villanueva-De La Cruz, Hortencia M. Saldaña-Cumpa, Mario J. Valladares-Garrido.

**Visualization:** Darwin A. León-Figueroa, Edwin A. Garcia-Vasquez, Milagros Diaz-Torres, Edwin Aguirre-Milachay, Jean Pierre Villanueva-De La Cruz, Hortencia M. Saldaña-Cumpa, Mario J. Valladares-Garrido.

**Writing – original draft:** Darwin A. León-Figueroa, Edwin A. Garcia-Vasquez, Milagros Diaz-Torres, Edwin Aguirre-Milachay, Jean Pierre Villanueva-De La Cruz, Hortencia M. Saldaña-Cumpa, Mario J. Valladares-Garrido.

**Writing – review & editing:** Darwin A. León-Figueroa, Edwin A. Garcia-Vasquez, Milagros Diaz-Torres, Edwin Aguirre-Milachay, Jean Pierre Villanueva-De La Cruz, Hortencia M. Saldaña-Cumpa, Mario J. Valladares-Garrido.

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
