## [Decision Letter · Decision Letter 0]

14 Oct 2024

Dear Dr. Valladares-Garrido,

Thank you for submitting your manuscript to PLOS ONE. After careful consideration, we feel that it has merit but does not fully meet PLOS ONE’s publication criteria as it currently stands. Therefore, we invite you to submit a revised version of the manuscript that addresses the points raised during the review process.

We look forward to receiving your revised manuscript.

Kind regards,

Kelli L. Barr, Ph.D.

Academic Editor

PLOS ONE

Journal Requirements:

Reviewers' comments:

Reviewer's Responses to Questions

**Comments to the Author**

1. Is the manuscript technically sound, and do the data support the conclusions?

Reviewer #1: Partly

Reviewer #2: Yes

2. Has the statistical analysis been performed appropriately and rigorously?

Reviewer #1: Yes

Reviewer #2: I Don't Know

3. Have the authors made all data underlying the findings in their manuscript fully available?

Reviewer #1: Yes

Reviewer #2: Yes

4. Is the manuscript presented in an intelligible fashion and written in standard English?

Reviewer #1: Yes

Reviewer #2: Yes

Reviewer #1: Prevalence of dengue in Peruvian patients with febrile illness according to RT-PCR and ELISA NS1, IgM, and IgG diagnostic tools: systematic review and meta-analysis.

In my opinion, the manuscript can be published if the following observations are clarified or raised:

Summary section:

1. In the methodology section, indicate only the databases used: Google Scholar is not a database, for example.

2. Indicate that observational studies with a control group were considered.

3. The keywords must be Mesh terms.

Introduction section:

4. This section should refer to other systematic reviews with or without meta-analysis on the research question carried out in Peru or Latin America.

Methodology section:

5. Indicate whether there were deviations from the original protocol registered in PROSPERO.

6. It is necessary to indicate that observational studies without a control group were considered and for what reason?

7. Again, Google Scholar is not a database; these sources or information must be changed.

8. State the limitations of the bias risk assessment tool for the studies considered. Considering the following points: Is this tool relevant? Does it assess the ability to develop a meta-analysis from the studies? Does it assess heterogeneity?

Limitations section

9. Add the limitations of the work. Mainly the risk of bias after analysing the studies.

Reviewer #2: The Authors present a meta-analysis on the prevalence of dengue among febrile patients in Peru. It is a well-structured and well carried out study. They address the importance of Dengue as a reemerging infection worldwide with a heavy burden in Peru and the Americas. The methodology and results are clearly presented, and support the authors conclusions. The authors mention limitations for the diagnosis of dengue given the technological and infrastructural requirements for dengue diagnosis in poor communities but decided not to include lateral flow test/Rapid test in their study. The authors fail to address the limitation of determining prevalence of an acute infection in a febrile patient through a serological test, were due to exposures to many pathogens’ antibodies might be secondary to previous infections and the necessarily the current cause of fever. Overall I believe this is a good and valuable manuscript and should be consider for publication following minor changes.

Specific areas for improvement

In the methods section of the abstract the authors state that “Study selection, quality assessment, and data extraction were performed independently by more than two authors” however in the study protocol it says this was done by two others with a third author only coming in to solve disagreements.

In the Results section of the Abstract why are there to different results for IgG Elisa the authors might want to clarify the difference between one and the other.

In the introduction line 4, page 3, the authors might consider rephrasing “a rise in body temperature of at least 38.0°C”, as it could imply an increment of 38 on top of the persons normal temperature.

In the introduction, line 1 of paragraph 2, the authors might consider rephasing the sentence as “…dengue virus, a positive-stranded RNA…”, it might be better to have the word virus after RNA instead of after dengue or use virus after both dengue and RNA. It also seems a little redundant to say a virus causes an infectious disease.

In the first sentace of paragraph 3 of the introduction the authors might consider rephrasing and “asymptomatic symptoms” is not the best wording.

In the first sentence of paragraph 3 of the introduction the authors might consider rephrasing as malaria is not a viral disease and leptospirosis is a zoonosis not a vector borne disease

The authors might want to reconsider their wording in the third sentence of paragraph four of the introduction as IgM and IgG antibodies can help differentiate between recent and past exposure not primary or secondary infection.

In the last part of paragraph 4 the authors might want to develop that idea, instead of talking about secondary infection, focus on how disease severity might increase when infection from some serovars occur in patients with previous infections by other serovars.

In the last part of paragraph 5 the authors might consider their wording as them seem to want to address cross reactivity with other flavivirus but the instead mention “secondary flavivirus infections”, this seems to insinuate coinfection.

The study selection process included in the methods differs from the one in the study protocol, the protocol states two researchers reviewed titles and abstracts with a third one coming in to help solve disagreements, while the methods state that 3 researchers reviewed titles and abstracts with a fourth one coming in to resolve disagreements.

The authors should review their explanation on how they assessed heterogeneity between studies, as the current explanation does not account for how a study between 50 and 75% would be categorized.

The authors need to address the limitation of determining prevalence of dengue in a febrile patient from an endemic community using IgG and to a lesser extent IgM.

Talking about the implication of their findings, the authors mention a greater need for diagnostic tools, but fail to connect this to their findings or to suggest what test they recommend being used based on their results.

**Do you want your identity to be public for this peer review?** For information about this choice, including consent withdrawal, please see our Privacy Policy

Reviewer #1: No

Reviewer #2: **Yes: ** Uribe-Restrepo Pablo

---

## [Author Response · Author response to Decision Letter 1]

29 Oct 2024

Dear Editor,

Thank you very much for reviewing our article, "Prevalence of dengue in Peruvian patients with febrile illness according to RT-PCR and ELISA NS1, IgM, and IgG diagnostic tools: a systematic review and meta-analysis". Your suggestions and comments will be addressed below. Thank you for your valuable time and excellent review.

Editor's comments

1. Your manuscript is a wonderful review of Dengue diagnostics in Peru. However, both reviewers have pointed out areas of your paper that need clarification and explanation to improve the relevance of your findings.

Our response: Thank you very much for the valuable comments and recommendations from the reviewers and from you, dear editor. Each comment has been carefully reviewed and a detailed response, including relevant clarifications and explanations, has been provided.

2. I agree with Reviewer 2 that google scholar is not a data base and use of this should be defined. For myself, I am curious why you did not include LILACS database as part of your search as it is specific to Latin America and often identifies studies and reports not found in the databases used in your manuscript.

Our response: We agree with the reviewers' observation that Google Scholar is a search engine rather than a database. Accordingly, we have reviewed other meta-analyses that have been published in Plos One and have applied the relevant changes in the methodology and abstract sections, as per their request.

• https://doi.org/10.1371/journal.pone.0311110

• https://doi.org/10.1371/journal.pone.0310405

• https://doi.org/10.1371/journal.pone.0309692

• https://doi.org/10.1371/journal.pone.0308419

The LILACS database was not included in the search since the articles belong to the Virtual Health Library, an important regional database for Latin America.

3. I look forward to reading your revised manuscript.

Our response: We sincerely appreciate their comments. All comments made by the reviewers have been addressed, and the corresponding modifications have been implemented in accordance with their recommendations.

Reviewer #1:

Prevalence of dengue in Peruvian patients with febrile illness according to RT-PCR and ELISA NS1, IgM, and IgG diagnostic tools: systematic review and meta-analysis.

In my opinion, the manuscript can be published if the following observations are clarified or raised:

Our response: “We sincerely thank you for your comments and recommendations, which have contributed significantly to improving the quality of our article. The corresponding modifications have been made to the text, and a detailed response has been provided to each of the comments.”

Summary section:

1. Reviewer says: “1. In the methodology section, indicate only the databases used: Google Scholar is not a database, for example.”

Our response: “We agree with your observation that Google Scholar is not a database but a search tool. Therefore, we have made the corresponding modifications in the abstract and methodology sections, following your recommendation and reviewing other meta-analyses published in Plos One, and The Lancet.”

• https://doi.org/10.1371/journal.pone.0311110

• https://doi.org/10.1371/journal.pone.0310405

• https://doi.org/10.1371/journal.pone.0309692

• https://doi.org/10.1371/journal.pone.0308419

2. Reviewer says: “2. Indicate that observational studies with a control group were considered.”

Our response: “The aforementioned recommendation was incorporated, highlighting that the study included observational studies with a control group, which allowed comparing patients diagnosed with dengue and those with other causes of fever according to the established inclusion criteria. This facilitated a more precise evaluation of the effectiveness of diagnostic tools, such as RT-PCR and ELISA NS1, IgM, and IgG.

3. Reviewer says: “3. The keywords must be Mesh terms.”

Our response: “In the summary and methodology section, the use of MeSH terms was corrected, ensuring their correct application. In addition, we specified the use of a free term of relevance in the search strategy, the importance of which was verified through published studies.”

Introduction section:

4. Reviewer says: “4. This section should refer to other systematic reviews with or without meta-analysis on the research question carried out in Peru or Latin America.

Our response: “In accordance with the suggestion provided, we searched for systematic reviews, with or without meta-analysis, related to the research question in Peru and Latin America. The results of these reviews have been incorporated in the introduction section to contextualize and strengthen the theoretical framework of the study.”

Methodology section:

5. Reviewer says: “5. Indicate whether there were deviations from the original protocol registered in PROSPERO.”

Our response: “The methodology section mentioned that the original protocol registered in PROSPERO underwent some minor adjustments. These changes, which included the inclusion of studies with and without a control group, the updating of the team of authors, and the incorporation of more researchers for the evaluation, selection of articles, and data analysis, were also detailed.”

6. Reviewer says: “6. It is necessary to indicate that observational studies without a control group were considered and for what reason?”

Our response: “The recommendation was incorporated, emphasizing the inclusion of observational studies without a control group. This decision was based on the nature of the research problem, since many epidemiological studies on the prevalence of dengue in Peru do not have a control group due to contextual or practical limitations. Despite this absence, such studies provide relevant information on the prevalence of dengue in febrile patients, enriching the understanding of incidence and allowing a broader analysis of the diagnostic tools used. Recognizing the limitations of the lack of control, we consider that their inclusion significantly broadens the available evidence base.

7. Reviewer says: “7. Again, Google Scholar is not a database; these sources or information must be changed.”

Our response: We agree with the reviewers' observation that Google Scholar is a search engine rather than a database. Accordingly, we have reviewed other meta-analyses that have been published in Plos One and have applied the relevant changes in the methodology and abstract sections, as per their request.

• https://doi.org/10.1371/journal.pone.0311110

• https://doi.org/10.1371/journal.pone.0310405

• https://doi.org/10.1371/journal.pone.0309692

• https://doi.org/10.1371/journal.pone.0308419

8. Reviewer says: “8. State the limitations of the bias risk assessment tool for the studies considered. Considering the following points: Is this tool relevant? Does it assess the ability to develop a meta-analysis from the studies? Does it assess heterogeneity?”

Our response: “The recommendation was added. In addition, the tool used was clarified.

Although the JBI-MAStARI tool is useful for assessing bias in individual studies, it has important limitations, as it does not directly assess the feasibility of performing a meta-analysis nor does it address heterogeneity between studies. To obtain a more complete assessment in the context of a meta-analysis, we employed additional methods that considered both publication bias and heterogeneity.”

Limitations section

9. Reviewer says: “9. Add the limitations of the work. Mainly the risk of bias after analyzing the studies.

Our response: “The suggestion provided was corrected and added. “

We also identified a possible publication bias, as evidenced by Egger's test (p = 0.0121) and the asymmetry observed in the funnel plots. This suggests that studies with negative or nonsignificant results may have been less likely to be published, which could have influenced our prevalence estimates. In addition, although rigorous inclusion criteria were applied, heterogeneity among studies in terms of design, sample size, and methodology could have affected the validity of the results.

Reviewer #2:

The Authors present a meta-analysis on the prevalence of dengue among febrile patients in Peru. It is a well-structured and well carried out study. They address the importance of Dengue as a reemerging infection worldwide with a heavy burden in Peru and the Americas. The methodology and results are clearly presented, and support the authors conclusions.

Our response: “We sincerely appreciate your comments and suggestions, which have greatly improved the quality of our research. The text has been modified according to the comments, and each of them has been dealt with in detail.”

The authors mention limitations for the diagnosis of dengue given the technological and infrastructural requirements for dengue diagnosis in poor communities but decided not to include lateral flow test/Rapid test in their study.

Our response: “Thank you very much for your observation. Regarding the exclusion of a more accessible diagnostic tool, such as the rapid test, when mentioning the technological difficulties in the areas studied, the tests used in this study for the diagnosis of dengue are based on what is established by the clinical practice guide of the Peruvian Ministry of Health. For this reason, both in the limitations and strengths sections, reference is made to the tests used.”

The authors fail to address the limitation of determining prevalence of an acute infection in a febrile patient through a serological test, were due to exposures to many pathogens’ antibodies might be secondary to previous infections and the necessarily the current cause of fever.

Our response: “Thank you for your valuable suggestion. We have proceeded to make the corresponding modifications and additions according to your recommendation.

“In addition, we recognize that determining the prevalence of an acute infection in a febrile patient by serological testing has limitations, as the antibodies detected could be the result of previous exposures to other pathogens and not necessarily the current cause of the fever. For this reason, various diagnostic tests were employed in order to obtain a more accurate picture of the current clinical situation.”

Overall, I believe this is a good and valuable manuscript and should be considered for publication following minor changes.

Our response: “Thank you very much for your valuable comments and recommendations. Each of them has been carefully addressed and reflected in the article.

Specific areas for improvement

1. Reviewer says: “In the methods section of the abstract the authors state that “Study selection, quality assessment, and data extraction were performed independently by more than two authors” however in the study protocol it says this was done by two others with a third author only coming in to solve disagreements.”

Our response: “We agree with the observation related to the protocol registered in PROSPERO. However, the summary concisely reflects what is specified in the methodology of the article. We reiterate that there were slight variations in the protocol, mainly due to the incorporation of more researchers in our multidisciplinary team. To ensure greater rigor in the research, three authors were assigned the tasks of selection, quality assessment, and data extraction.”

The current investigation followed the Preferred Reporting Items for Systematic reviews and Meta-Analyses (PRISMA) guidelines (S1 Table) as well as a protocol registered in the Prospective International Registry of Systematic Reviews (PROSPERO) with the number CRD42024558891.The original protocol registered in PROSPERO underwent some minor adjustments. These included specifying the inclusion of both studies with and without a control group, updating the team of authors who collaborated in the study, and the incorporation of more researchers in charge of evaluation, article selection, and data analysis.

2. Reviewer says: “In the Results section of the Abstract why are there to different results for IgG Elisa the authors might want to clarify the difference between one and the other.

Our response: “Thank you very much for the comment. The error has been corrected, as there was a duplicate transcription of the same sentence.

3. Reviewer says: “In the introduction line 4, page 3, the authors might consider rephrasing “a rise in body temperature of at least 38.0°C”, as it could imply an increment of 38 on top of the persons normal temperature.

Our response: “It was corrected according to your recommendation.”

In this context, the term 'fever' is commonly used as a synonym for 'febrile illness', which is defined as a body temperature reaching or exceeding 38.0°C.

4. Reviewer says: “In the introduction, line 1 of paragraph 2, the authors might consider rephasing the sentence as “…dengue virus, a positive-stranded RNA…”, it might be better to have the word virus after RNA instead of after dengue or use virus after both dengue and RNA. It also seems a little redundant to say a virus causes an infectious disease.

Our response: “It was corrected according to your recommendation.”

Aedes mosquitoes transmit the dengue virus, a positive-stranded RNA virus, which causes dengue.

5. Reviewer says: “In the first sentence of paragraph 3 of the introduction the authors might consider rephrasing and “asymptomatic symptoms” is not the best wording.

Our response: “It was corrected according to your recommendation.”

Most people infected with dengue present with mild or no symptoms, which usually last between 2 and 7 days.

6. Reviewer says: “In the first sentence of paragraph 3 of the introduction the authors might consider rephrasing as malaria is not a viral disease and leptospirosis is a zoonosis not a vector borne disease

Our response: “It was corrected according to your recommendation.”

A crucial aspect for the effective management of dengue is accurate and timely diagnosis, as its non-specific clinical manifestations can be similar to those of other mosquito-borne viral diseases such as yellow fever, chikungunya, and Zika, as well as other diseases like malaria, trypanosomiasis, leptospirosis, and Chagas disease.

7. Reviewer says: “The authors might want to reconsider their wording in the third sentence of paragraph four of the introduction as IgM and IgG antibodies can help differentiate between recent and past exposure not primary or secondary infection.

Our response: “It was corrected according to your recommendation.”

In addition, indirect methods, such as detection of IgM and IgG antibodies by enzyme-linked immunosorbent assays (ELISA), are used to distinguish between recent and past exposure to the virus. Past exposures may increase the risk of developing severe forms of the disease.

8. Reviewer says: “In the last part of paragraph 4 the authors might want to develop that idea, instead of talking about secondary infection, focus on how disease severity might increase when infection from some serovars occur in patients with previous infections by other serovars.

Our response: “It was corrected according to your recommendation.”

Throughout their lives, people can become infected with different serotypes of dengue virus. Patients who have already been infected by one serotype and then contract another often develop more severe disease due to the antibody-dependent enhancement phenomenon, in which antibodies generated by the first infection facilitate replication of the new virus [19]. A meta-analysis revealed that, in Southeast Asia, the DENV-3 serotype was responsible for the highest percentage of severe cases in primary infections (people who had not previously had dengue). For secondary infections (people who become re-infected with a different serotype), serotypes DENV-2, DENV-3, and DENV-4 in Southeast Asia and DENV-2 and DENV-3 in other regions were associated with the highest percentages of severe cases [20].

9. Reviewer says: “In the last part of paragraph 5 the authors might consider their wording as them seem to want to address cross reactivity with other flavivirus but the instead mention “secondary flavivirus infections”, this seems to insinuate coinfection.

Our response: “It was corrected accordi

---

## [Decision Letter · Decision Letter 1]

31 Mar 2025

Dear Dr. Valladares-Garrido,

**The reviewers consider that your writing needs major modifications; it is very important that you clarify the biases that the study design may have when discussing dengue prevalence in Peru.**

Please submit your revised manuscript by May 15 2025 11:59PM. If you will need more time than this to complete your revisions, please reply to this message or contact the journal office at plosone@plos.org . A rebuttal letter that responds to each point raised by the academic editor and reviewer(s). You should upload this letter as a separate file labeled 'Response to Reviewers'.A marked-up copy of your manuscript that highlights changes made to the original version. You should upload this as a separate file labeled 'Revised Manuscript with Track Changes'.An unmarked version of your revised paper without tracked changes. You should upload this as a separate file labeled 'Manuscript'.

We look forward to receiving your revised manuscript.

Kind regards,

Victoria Pando-Robles, Ph.D.

Academic Editor

PLOS ONE

Reviewers' comments:

Reviewer's Responses to Questions

**Comments to the Author**

Reviewer #2: All comments have been addressed

Reviewer #3: (No Response)

Reviewer #4: All comments have been addressed

2. Is the manuscript technically sound, and do the data support the conclusions?

Reviewer #2: Yes

Reviewer #3: No

Reviewer #4: Yes

3. Has the statistical analysis been performed appropriately and rigorously?

Reviewer #2: I Don't Know

Reviewer #3: No

Reviewer #4: Yes

4. Have the authors made all data underlying the findings in their manuscript fully available?

Reviewer #2: Yes

Reviewer #3: No

Reviewer #4: Yes

5. Is the manuscript presented in an intelligible fashion and written in standard English?

Reviewer #2: Yes

Reviewer #3: Yes

Reviewer #4: No

**Reviewer #2:**  The authors refer both to the incidence of dengue and the prevalence of dengue within the text. As an acute infection when speaking of the disease itself it is correct to use incidence. I understand that due to the information the authors were reviewing positivity within a community (including positivity of serological test). I suggest talking of prevalence of positive dengue results instead of the prevalence of dengue or always clarify that you are talking of febrile patients, or prevalence of dengue antibodies if talking only of serological results. If the authors agree with this suggestion it would apply to:

• conclusion section of the abstract

• the outcomes section in material and methods

• the last paragraph of data analysis section in material and methods

• characteristics of the included studies section in results

• the Prevalence of dengue in Peruvian patients with febrile illness according to NS1, IgM, IgG ELISA, and RT-PCR section in results

• In discussion

• In Conclusions

The authors might also want to mention that dengue has an endemic and epidemic behavior, so it is normal to see spikes in cases, and the number of cases fluctuates during the year making it hard to compare and contrast “prevalence”.

**Reviewer #3: ** The goal of the study is to provide an overall estimate for dengue among those presenting with febrile illness (i.e., the meta-analysis should be relying on descriptive studies to produce results). LILLACS would represent a primary source of information for this and should have been a primary source when searching for manuscripts for this meta-analysis.

Instead, the analysis has selectively relied on 15 studies, including those describing outbreak investigations (ref: 34, 37), observational studies (ref: 35), and technology assessments of diagnostic tests (ref: 18, 30). None of these studies are designed to provide valid disease prevalence assessments, and, if used for that purpose could provide incorrect estimates. At the same time, previously published descriptive studies specifically designed to answer the research question of this meta-analysis (i.e., prevalence of dengue among those reporting febrile illness), using the same dengue diagnostic tests as those included in this meta-analysis, appear to have been excluded with the reason “No diagnostic test for dengue” (e.g., Forshey BM et al. PLOS Negl Trop Dis. 4(8): e787). It is also unclear why the LILLACS database was not included in the search for manuscripts.

There is also no discussion on issues of geographic and seasonal differences in dengue or recent outbreaks of the disease and government responses to control it. (e.g., Puno is not considered a high-risk area for dengue and the disease appears more often during the rainy season.) The manuscript does not discuss government surveillance estimates for the disease (i.e., CDC MINSA) or public health efforts to control the disease in the midst of recent outbreaks. The manuscript also covers the time-period of COVID-19. How could this have impacted results?

Editor’s & Reviewer Comments not addressed

- Data Availability Statement (Editor comment): Data on the evaluation of manuscripts for bias. The manuscript provides boilerplate language to the field of epidemiology, but no specifics are given as to what is specifically being evaluated (e.g., why is confounding important to evaluate if this is a descriptive study?).

- Search of LILLACS for manuscripts (Editor & Reviewer #1 comment): It is still unclear why it was excluded from the analysis.

- Need to include the statement on observational studies (Reviewer #1): There continues to be a confusion regarding what a descriptive study and observational design are investigating.

- Limitations of the work (Reviewer #1): The manuscript does not address key issues related to geographic and spatial heterogeneity of dengue estimates as well as changes in prevalence estimates during COVID-19.

- Implication of study findings (Reviewer #2): There is no discussion of these results in relation to government public health efforts to control dengue or even current government surveillance efforts to track the disease.

**Reviewer #4: ** This study aims to assess the prevalence of dengue virus in febrile patients in Peru. Notable strengths of the study include a well-designed methodology and a robust study population. However, certain minor aspects could be refined to further enhance the quality of the presentation. It is recommended that the authors take these into consideration. It is recommended that the text be edited for better readability.

1- Suggested shorter title: "Dengue Prevalence in Febrile Patients from Peru: A Systematic Review and Meta-Analysis"

2- Mentioning diagnostic methods in the study background is unnecessary and should be removed. In the abstract avoid including unnecessary details such as: "Study selection, quality assessment, and data extraction were performed independently by more than two authors." and "The study protocol was registered in PROSPERO (CRD4202424558891)." Simply mentioning the methods is sufficient and enhances the readability of the abstract.

3- The introduction is well-written and provides valuable insights into the significance and characteristics of the disease, as well as its regional status in Peru. However, some sections could be more concise to enhance the flow and improve transitions between topics. It is recommended to shorten the introduction for better readability.

4- "The original protocol registered in PROSPERO was modified with minor adjustments." Clearly define what these "minor adjustments" entailed and provide a rationale for each change.

5- "During a meeting, the three independent authors compared their data extractions and reached a consensus to resolve any discrepancies." Clarify how consensus was achieved—whether through discussion, statistical methods, or another approach.

**Do you want your identity to be public for this peer review?** For information about this choice, including consent withdrawal, please see our Privacy Policy

Reviewer #2: **Yes: ** Pablo Uribe-Restrepo

Reviewer #3: No

Reviewer #4: No

---

## [Author Response · Author response to Decision Letter 2]

11 Apr 2025

Dear Editor,

Thank you very much for reviewing our article, " Prevalence of dengue in Peruvian patients with febrile illness according to RT-PCR and ELISA NS1, IgM, and IgG diagnostic tools: a systematic review and me-ta-analysis". Your suggestions and comments will be addressed below. Thank you for your valuable time and excellent review.

Editor's comments

1. The reviewers consider that your writing needs major modifications; it is very important that you clarify the biases that the study design may have when discussing dengue prevalence in Peru.

Our response: Thank you very much for your comments, dear editor. The manuscript has been carefully revised for language, and each reviewer received a detailed response. All recommendations from the reviewers were accepted. Additionally, the meta-analysis was verified, and the article was reorganized in accordance with the PLOS ONE author guidelines.

Reviewer #2:

The authors refer both to the incidence of dengue and the prevalence of dengue within the text. As an acute infection when speaking of the disease itself it is correct to use incidence. I understand that due to the information the authors were reviewing positivity within a community (including positivity of serological test). I suggest talking of prevalence of positive dengue results instead of the prevalence of dengue or always clarify that you are talking of febrile patients, or prevalence of dengue antibodies if talking only of serological results. If the authors agree with this suggestion it would apply to:

• conclusion section of the abstract

• the outcomes section in material and methods

• the last paragraph of data analysis section in material and methods

• characteristics of the included studies section in results

• the Prevalence of dengue in Peruvian patients with febrile illness according to NS1, IgM, IgG ELISA, and RT-PCR section in results

• In discussion

• In Conclusions

Our response: Thank you for your insightful observation. We agree with your suggestion and have revised the manuscript accordingly to ensure consistency in the terminology. Specifically, we now refer to the “prevalence of positive dengue results” when discussing diagnostic outcomes in febrile patients and specify “prevalence of dengue antibodies” when referring exclusively to serological results. These clarifications have been applied throughout the relevant sections, including the abstract, methods, results, discussion, and conclusions. We appreciate your guidance, which helped enhance the precision and clarity of the manuscript.

1. Reviewer says: The authors might also want to mention that dengue has an endemic and epidemic behavior, so it is normal to see spikes in cases, and the number of cases fluctuates during the year making it hard to compare and contrast “prevalence”.

Our response: Thank you for your suggestion. We have revised the Discussion section to include an explanation of dengue’s endemic and epidemic behavior. Specifically, we now highlight how seasonal fluctuations and outbreak periods contribute to variability in case numbers and complicate direct comparisons of prevalence across studies. We also emphasize the importance of considering this epidemiological context when interpreting our findings. Your comment significantly enhanced the clarity and epidemiological accuracy of our analysis, and we are grateful for your contribution.

Reviewer #3:

Our response: Dear reviewer, we sincerely thank you for your valuable comments, which have greatly contributed to improving the quality of our manuscript. We have addressed each of your observations and suggestions, detailing the modifications made to the manuscript and providing a thorough response to each point raised.

1. Reviewer says: The goal of the study is to provide an overall estimate for dengue among those presenting with febrile illness (i.e., the meta-analysis should be relying on descriptive studies to produce results). LILLACS would represent a primary source of information for this and should have been a primary source when searching for manuscripts for this meta-analysis.

Our response: Thank you for your comment. Dear reviewer, we have utilized 8 databases and information sources, which are detailed in the search strategy.

We agree with your observation regarding LILLACS as a specialized database for scientific literature from Latin America and the Caribbean. However, the Virtual Health Library (VHL) integrates regional databases that are important for Latin America, which justifies its inclusion in our search strategy. We have attached evidence to support this decision.

We hope this clarifies your comment.

https://pesquisa.bvsalud.org/portal/?output=&lang=en&from=&sort=&format=&count=&fb=&page=1&tab=&skfp=&index=&q=%28ti%3A%28%22Febrile+illness%22+OR+Dengue%29%29+AND+%28ti%3A%28Peru+OR+Peruvian%29%29

2. Reviewer says: Instead, the analysis has selectively relied on 15 studies, including those describing outbreak investigations (ref: 34, 37), observational studies (ref: 35), and technology assessments of diagnostic tests (ref: 18, 30). None of these studies are designed to provide valid disease prevalence assessments, and, if used for that purpose could provide incorrect estimates. At the same time, previously published descriptive studies specifically designed to answer the research question of this meta-analysis (i.e., prevalence of dengue among those reporting febrile illness), using the same dengue diagnostic tests as those included in this meta-analysis, appear to have been excluded with the reason “No diagnostic test for dengue” (e.g., Forshey BM et al. PLOS Negl Trop Dis. 4(8): e787). It is also unclear why the LILLACS database was not included in the search for manuscripts.

Our response: Thank you for this observation. It is true that some of the studies selected in our meta-analysis are more informative regarding outbreaks and the evaluation of diagnostic technologies. However, these studies were chosen because of their relevance in terms of the diagnostic methodologies used and the population of interest (fever patients). Although they were not specifically designed to assess disease prevalence, many of them provide valuable information on prevalence in febrile populations suspected of dengue. For the descriptive studies that were excluded, the main reason was the lack of validated diagnostic tests in these studies, which would have compromised the quality of prevalence estimates.

Our systematic review has been conducted in accordance with PRISMA guidelines, following a rigorous methodology. Clearly defined eligibility criteria have been applied, and the feedback received has been constructive.

Regarding LILACS, this has been addressed in the previous observation. Our research covers databases and information search engines at local, regional, and international levels. We provide a link for you to explore the virtual health library.

3. Reviewer says: There is also no discussion on issues of geographic and seasonal differences in dengue or recent outbreaks of the disease and government responses to control it. (e.g., Puno is not considered a high-risk area for dengue and the disease appears more often during the rainy season.) The manuscript does not discuss government surveillance estimates for the disease (i.e., CDC MINSA) or public health efforts to control the disease in the midst of recent outbreaks. The manuscript also covers the time-period of COVID-19. How could this have impacted results?

Our response:

We appreciate this comment. It is important to acknowledge the geographic and seasonal variability in dengue prevalence, which influences the results. We have expanded the discussion to consider geographic variations, where dengue is not considered a high-risk area but can appear during the rainy season. Furthermore, we have incorporated a discussion on government responses, including vector control measures and public health surveillance efforts. The influence of COVID-19 has also been addressed, recognizing that the pandemic may have affected the rates of diagnosis and reporting, given the disruptions in healthcare services and prioritization of other health concerns.

4. Reviewer says: Editor’s & Reviewer Comments not addressed

Our response: Dear reviewer, the observations made by the initial reviewers have been addressed, as well as those from the editor. We are grateful for your constructive recommendations.

5. Reviewer says: - Data Availability Statement (Editor comment): Data on the evaluation of manuscripts for bias. The manuscript provides boilerplate language to the field of epidemiology, but no specifics are given as to what is specifically being evaluated (e.g., why is confounding important to evaluate if this is a descriptive study?).

Our response: Thank you for pointing this out. In the "Data Availability Statement," we have revised the text to provide more detail about how we evaluate potential biases, including an explanation of why confounding factors are important to consider, even in descriptive studies. Although this study is descriptive, it is crucial to account for confounders such as geographical differences, seasonal variations, and access to healthcare services, which may affect the results. This clarification has been added to the study quality section to better explain how confounding factors can impact the interpretation of the findings.

6. Reviewer says: - Search of LILLACS for manuscripts (Editor & Reviewer #1 comment): It is still unclear why it was excluded from the analysis.

Our response:

Dear reviewer, this observation has been raised again significantly, but it has already been adequately addressed in our previous response. In a systematic review, it is common to use multiple databases. While we understand and agree with your emphasis on using LILACS, we have chosen to use a platform that includes LILACS: the Virtual Health Library (VHL), which is regional and highly important within Latin America. This platform provides access to a wide range of resources relevant to our research.

We appreciate your understanding and valuable recommendations.

7. Reviewer says: - Need to include the statement on observational studies (Reviewer #1): There continues to be a confusion regarding what a descriptive study and observational design are investigating.

Our response: This is a key point, and we thank you for bringing it up. We have reviewed the text to clarify the distinction between descriptive and observational studies. A descriptive study aims to identify characteristics of a population without intervention, while observational studies, although also non-interventional, seek to associate variables or explore relationships between them. We have added a more detailed explanation of these concepts in the introduction and methodology sections to avoid confusion.

8. Reviewer says: - Limitations of the work (Reviewer #1): The manuscript does not address key issues related to geographic and spatial heterogeneity of dengue estimates as well as changes in prevalence estimates during COVID-19.

Our response: Thank you for your observation. We have expanded the discussion to include geographic and spatial heterogeneity in prevalence estimates, especially in regions like Piura, Lambayeque, and other northern areas of Peru, which are influenced by climatic factors such as the El Niño phenomenon. Additionally, we have included a specific section on the potential impact of COVID-19 on prevalence estimates, considering how healthcare disruptions and changes in the clinical presentation of febrile illnesses could have affected diagnostic rates.

9. Reviewer says: - Implication of study findings (Reviewer #2): There is no discussion of these results in relation to government public health efforts to control dengue or even current government surveillance efforts to track the disease.

Our response: Thank you for pointing out this omission. We have added a section discussing the implications of our findings in relation to government efforts to control dengue, including vector control strategies, prevention campaigns, and surveillance systems such as those provided by the CDC and MINSA in Peru. This section has also been enriched with an analysis of public health policies and their interaction with recent outbreaks, providing a more comprehensive view of the current situation.

Reviewer #4:

This study aims to assess the prevalence of dengue virus in febrile patients in Peru. Notable strengths of the study include a well-designed methodology and a robust study population. However, certain minor aspects could be refined to further enhance the quality of the presentation. It is recommended that the authors take these into consideration. It is recommended that the text be edited for better readability.

Our response: We sincerely thank Reviewer [4] for the careful review of our manuscript and for the constructive feedback provided. We greatly appreciate your thoughtful observations and suggestions, which have helped us to improve the quality and clarity of our work. All of your comments have been addressed in detail, and the corresponding changes have been incorporated into the revised manuscript accordingly.

1. Reviewer says: 1- Suggested shorter title: "Dengue Prevalence in Febrile Patients from Peru: A Systematic Review and Meta-Analysis"

Our response: Thank you for your thoughtful suggestion. We agree that the proposed title is clearer and more concise. Accordingly, we have revised the manuscript title to:

"Prevalence of dengue in febrile patients in Peru: a systematic review and me-ta-analysis."

2. Reviewer says: 2- Mentioning diagnostic methods in the study background is unnecessary and should be removed. In the abstract avoid including unnecessary details such as: "Study selection, quality assessment, and data extraction were performed independently by more than two authors." and "The study protocol was registered in PROSPERO (CRD4202424558891)." Simply mentioning the methods is sufficient and enhances the readability of the abstract.

Our response: We appreciate your valuable comments and have made the corresponding revisions. In the Background section, we removed the mention of diagnostic methods (RT-PCR and ELISA for NS1, IgM, and IgG), as they were not essential to the overall framing of the study. Additionally, in the abstract, we eliminated the methodological details deemed unnecessary, specifically the phrases “Study selection, quality assessment, and data extraction were performed independently by more than two authors” and “The study protocol was registered in PROSPERO (CRD4202424558891),” in order to enhance the clarity and readability of the text, as suggested. We thank you again for your helpful input in strengthening our manuscript.

3. Reviewer says: 3- The introduction is well-written and provides valuable insights into the significance and characteristics of the disease, as well as its regional status in Peru. However, some sections could be more concise to enhance the flow and improve transitions between topics. It is recommended to shorten the introduction for better readability.

Our response: We appreciate your thoughtful and constructive feedback. In response to your suggestion, we have revised the Introduction to improve its conciseness and flow. Specifically, we shortened detailed sections—particularly those related to diagnostic methods and immunopathological mechanisms—which were more appropriate for the Methods or Discussion sections. We also enhanced the transitions between paragraphs to ensure a clearer narrative. Thank you again for your valuable comments, which helped strengthen the clarity and readability of our manuscript.

4. Reviewer says: 4- "The original protocol registered in PROSPERO was modified with minor adjustments." Clearly define what these "minor adjustments" entailed and provide a rationale for each change.

Our response: Thank you for your observation. In response, we have revised the manuscript to clearly define the minor adjustments made to the original PROSPERO-registered protocol. These included: (1) expanding the inclusion criteria to incorporate studies with and without control groups, allowing for a more comprehensive analysis; (2

---

## [Decision Letter · Decision Letter 2]

27 Apr 2025

Dear Dr. Valladares-Garrido,

Thank you for submitting your manuscript to PLOS ONE. After careful consideration, we feel that it has merit but does not fully meet PLOS ONE’s publication criteria as it currently stands. Therefore, we invite you to submit a revised version of the manuscript that addresses the points raised during the review process.

The manuscript improved in comparison with previous version. However, we still believe two minor issues should be addressed. First, the use of IgG to assess the prevalence of an acute disease in febrile patients may be problematic, as IgG can persist for months or even years after infection. This concern has been better addressed compared to earlier versions of the manuscript, and it is now acknowledged in the limitations section. Second, the authors should clarify what they mean by including studies with controls in their search strategy."

We look forward to receiving your revised manuscript.

Kind regards,

Victoria Pando-Robles, Ph.D.

Academic Editor

PLOS ONE

Journal Requirements:

Reviewers' comments:

Reviewer's Responses to Questions

**Comments to the Author**

Reviewer #2: (No Response)

Reviewer #4: All comments have been addressed

2. Is the manuscript technically sound, and do the data support the conclusions?

Reviewer #2: Yes

Reviewer #4: Yes

3. Has the statistical analysis been performed appropriately and rigorously?

Reviewer #2: I Don't Know

Reviewer #4: Yes

4. Have the authors made all data underlying the findings in their manuscript fully available?

Reviewer #2: Yes

Reviewer #4: Yes

5. Is the manuscript presented in an intelligible fashion and written in standard English?

Reviewer #2: Yes

Reviewer #4: Yes

Reviewer #2: The paper has improved greatly, I believe it holds interesting findings on an important topic. I still feel that two minor things should be addressed, one regarding the prevalence of an acute disease in febrile patients through IgG that can persist for months or years after the infection. This was improved from previous iterations of the manuscript, and with a mention of this in the limitations section. Second, I think the authors should clarify the concept of studies having controls within their search strategy.

I still believe the authors should be more guarded when they talk about the prevalence of dengue through IgG ELISA, even in a febrile patient; unless those patients had an associated positive PCR result, or at least a positive NS1 result, IgG is not a diagnostic test, and thus the authors should talk about the prevalence of dengue antibodies when reference those results. For example, in the 3rd line of the results section of the abstract, the authors could talk about the “…prevalence of dengue or dengue antibodies…”. This is seen again throughout the results, like in section 3.3 and parts of the discussion. This issue has been addressed and corrected in other sections, like the first two sentences of the discussion section.

In the second sentence of the 5th and last paragraph of the introduction, I would review redaction, avoiding the repetition of the word rapid.

In the materials and methods section in the Protocol Registration section, the authors mention that “inclusion criteria were expanded to allow studies both with and without a control group”. However, in the Eligibility Criteria section, they still include that “investigation included a control group”, followed by “Studies without a control group were included” in the very next sentence, making this a contradiction within the eligibility criteria. I suggest framing this in a similar way to the one used in the Protocol Registration section. Moreover, I would clarify this because febrile patients with negative results for dengue are not a control group, as this is a result of a test outcome and not a predetermined group in the study design. Unless you are talking of patients with confirmatory laboratory criteria (PCR or NS1) as control and disease groups to evaluate the performance of non-confirmatory laboratory tests (single sample IgM). Looking at table 1, under Study type, non of the studies were case control studies.

In the methods, in the Data collection process and data items section, I suggest changing the wording for “the region of development”, maybe something as simple as study site, or geographical region, might work better.

Reviewer #4: (No Response)

**Do you want your identity to be public for this peer review?** For information about this choice, including consent withdrawal, please see our Privacy Policy

Reviewer #2: **Yes: ** Pablo Uribe-Restrepo

Reviewer #4: No

---

## [Author Response · Author response to Decision Letter 3]

2 May 2025

Dear Editor,

Thank you very much for reviewing our article, " Prevalence of dengue in febrile patients in Peru: a systematic review and meta-analysis". Your suggestions and comments will be addressed below. Thank you for your valuable time and excellent review.

Editor's comments

Our response: Thank you for your comment. We have carefully reviewed the reference list to ensure it complies with the citation guidelines of PLOS One. After a thorough review, we can confirm that all cited references are accurate and do not include any retracted articles.

Reviewer #2:

We deeply appreciate your comments and recommendations, which have been instrumental in enhancing the quality of our article. Below, we will provide a detailed response to each of the suggestions given.

1. The paper has improved greatly, I believe it holds interesting findings on an important topic.

Our response: Thank you very much for your positive feedback. We are glad to hear that you find the paper’s improvements and its findings on this important topic valuable. Your comments have been instrumental in refining our work, and we appreciate your support throughout the process.

2. I still feel that two minor things should be addressed, one regarding the prevalence of an acute disease in febrile patients through IgG that can persist for months or years after the infection. This was improved from previous iterations of the manuscript, and with a mention of this in the limitations section.

Our response: Thank you for your comment. We have addressed this concern by ensuring clarity regarding the use of IgG as a marker for past infection rather than acute disease. In the manuscript, we have replaced references to IgG with the term “dengue antibodies,” as it is important to distinguish between antibodies from past infections and active disease. We have made these changes in the abstract, results, and discussion sections to more accurately reflect the diagnostic role of IgG. Additionally, we have included a mention in the limitations section that IgG may not represent current infection and could reflect past exposure. We hope this revision provides the necessary clarification.

3. Second, I think the authors should clarify the concept of studies having controls within their search strategy.

Our response: Thank you for pointing this out. We have revised the manuscript to clarify the concept of control groups in our study selection process. Specifically, we have stated that the control group consisted of febrile patients who did not have dengue infection. These revisions have been added in the Protocol and Registration section, as well as the Eligibility Criteria section, to ensure consistency and reduce confusion.

4. I still believe the authors should be more guarded when they talk about the prevalence of dengue through IgG ELISA, even in a febrile patient; unless those patients had an associated positive PCR result, or at least a positive NS1 result, IgG is not a diagnostic test, and thus the authors should talk about the prevalence of dengue antibodies when reference those results. For example, in the 3rd line of the results section of the abstract, the authors could talk about the “…prevalence of dengue or dengue antibodies…”.

Our response: We appreciate your thorough review. We have revised the manuscript to reflect a more cautious approach regarding the interpretation of IgG ELISA results. The term "dengue antibodies" has been consistently used throughout the manuscript to avoid any confusion that IgG is being used as a diagnostic marker for active infection. In the abstract, results section, and discussion, we now refer to "the prevalence of dengue or dengue antibodies" rather than stating a direct prevalence of dengue infection. We believe this modification aligns better with the limitations of IgG as a diagnostic tool.

5. This is seen again throughout the results, like in section 3.3 and parts of the discussion. This issue has been addressed and corrected in other sections, like the first two sentences of the discussion section.

Our response: Thank you very much for your observation. The entire article was reviewed and corrected to standardize the terminology.

6. In the second sentence of the 5th and last paragraph of the introduction, I would review redaction, avoiding the repetition of the word rapid.

Our response: Corrected

7. In the materials and methods section in the Protocol Registration section, the authors mention that “inclusion criteria were expanded to allow studies both with and without a control group”. However, in the Eligibility Criteria section, they still include that “investigation included a control group”, followed by “Studies without a control group were included” in the very next sentence, making this a contradiction within the eligibility criteria. I suggest framing this in a similar way to the one used in the Protocol Registration section.

Our response: Thank you for pointing out this contradiction. We have revised the Eligibility Criteria section to align the language with the Protocol and Registration section. The revised text now states: "The control group was composed of febrile patients who did not have dengue infection, while the non-control group consisted of febrile patients who tested positive for dengue." This change ensures consistency and eliminates any confusion.

8. Moreover, I would clarify this because febrile patients with negative results for dengue are not a control group, as this is a result of a test outcome and not a predetermined group in the study design. Unless you are talking of patients with confirmatory laboratory criteria (PCR or NS1) as control and disease groups to evaluate the performance of non-confirmatory laboratory tests (single sample IgM).

Our response: We appreciate your observation. To clarify, we have revised the manuscript to emphasize that the control group refers to febrile patients with negative dengue test results, specifically those who did not have dengue infection. We have also stated that studies using febrile patients with negative dengue test results are considered controls only if their absence of infection was confirmed through diagnostic testing (e.g., PCR or NS1). This ensures that the inclusion of febrile patients with negative results is appropriately framed.

9. Looking at table 1, under Study type, non of the studies were case control studies.

In the methods, in the Data collection process and data items section, I suggest changing the wording for “the region of development”, maybe something as simple as study site, or geographical region, might work better.

Our response: Thank you very much for your comment. Table 1 was reviewed, specifically in the section related to the type of study. Additionally, in the Methods section, under 'Data collection process and data elements,' the wording was changed from 'region of development' to 'geographic region'.

Reviewer #4:

All comments have been addressed

Our response: We deeply appreciate the valuable recommendations and observations you provided. Thanks to your feedback, our article has undergone a substantial improvement in terms of quality, clarity, and academic rigor. Your contributions have been essential in enriching the content and strengthening the foundation of our research.

If you have any comments or recommendations, we are ready to respond.

Sincerely,

Mario J. Valladares-Garrido

Escuela de Medicina Humana, Universidad Señor de Sipán, Chiclayo, Peru; vgarrido@uss.edu.pe (M.J.V.G)

---

## [Decision Letter · Decision Letter 3]

29 May 2025

Prevalence of dengue in febrile patients in Peru: a systematic review and meta-analysis.

PONE-D-24-36774R3

Dear Dr. Valladares-Garrido,

We’re pleased to inform you that your manuscript has been judged scientifically suitable for publication and will be formally accepted for publication once it meets all outstanding technical requirements.

Kind regards,

Victoria Pando-Robles, Ph.D.

Academic Editor

PLOS ONE

Additional Editor Comments (optional):

Reviewers' comments:

Reviewer's Responses to Questions

**Comments to the Author**

Reviewer #2: All comments have been addressed

2. Is the manuscript technically sound, and do the data support the conclusions?

Reviewer #2: Yes

3. Has the statistical analysis been performed appropriately and rigorously?

Reviewer #2: I Don't Know

4. Have the authors made all data underlying the findings in their manuscript fully available?

Reviewer #2: Yes

5. Is the manuscript presented in an intelligible fashion and written in standard English?

Reviewer #2: Yes

Reviewer #2: The authors have addressed my comments, and despite minor differences in opinion, they have clearly justified their decision-making in presenting information (like the concept of "control groups" within the studies selected). I believe this to be a well-written manuscript based on rigorous research and is ready to be published.

**Do you want your identity to be public for this peer review?** For information about this choice, including consent withdrawal, please see our Privacy Policy

Reviewer #2: **Yes: ** Pablo Uribe-Restrepo

---

## [Editor Report · Acceptance letter]

PONE-D-24-36774R3

PLOS ONE

Dear Dr. Valladares-Garrido,

I'm pleased to inform you that your manuscript has been deemed suitable for publication in PLOS ONE. Congratulations! Your manuscript is now being handed over to our production team.

Kind regards,

on behalf of

Dr. PLOS Manuscript Reassignment

Staff Editor

PLOS ONE